# Drinking and swimming around waterways: The role of alcohol, sensation-seeking, peer influence and risk in young people

J. E. Leavy[1‡]*, M. Della Bona[1☉], M. Abercromby[2☉], G. Crawford[1‡]

**1** Collaboration for Evidence, Research and Impact in Public Health, Curtin University, Perth, Western Australia, Australia, **2** Royal Life Saving Society Western Australia, Perth, Western Australia, Australia

☉ These authors contributed equally to this work.
‡ JEL and GC are joint senior authors on this work.
* j.leavy@curtin.edu.au

**Data Availability Statement:** Data cannot be shared publicly because of ethical and contractual requirements between Curtin University, Royal Life Saving Society Western Australia and Department

## Abstract

The role of individual and sociocultural factors contributing to drowning risk for young adults is complex and poorly understood. This study examined the relationship between behaviour in and around waterways and: 1) alcohol consumption; 2) resistance to peer influence; 3) sensation-seeking; 4) perception of risk among people aged 15–24 in Western Australia. A cross-sectional online survey was conducted at three time-points with a convenience sample. Predictor variables included: Alcohol Use Disorder Identification Test Consumption (AUDIT_C); Resistance to Peer Influence; Brief Sensation Seeking scale; Benthin's Perception of risk. Pearson chi-squared tests determined the association between demographic and predictor variables. Logistic regression explored influence of potential predictor variables on behaviour in and around water. The final sample (n = 730) participants, consisted of females (n = 537, 74.5%), metropolitan dwelling (n = 616, 84.4%), and attended university (n = 410, 56.9%). Significant associations were found for those who swum after drinking alcohol compared with those that had not by age, gender, education. For every 1-unit increase in AUDIT-C participants were 60% more likely to swim after drinking (OR 95% CI 1.60 1.44–1.78). Participants who considered an adverse event serious were 15% less likely to have swum after drinking alcohol (OR 0.85 95% CI 0.73–0.99). The complex relationship between social participation in activities in and around waterways, higher drowning rates, propensity for risk, and the meaning young adults attach to risk locations and practices present unique challenges for drowning prevention research. Findings should be used to improve the awareness and education components of future youth water safety strategies in high-income settings.

## Introduction

Adolescents and young people drowning is a preventable public health problem associated with a significant social burden [1, 2]. Drowning is among the ten leading causes of death for people aged between one and 24 years in every region of the world, with around one half of all

of Health, Western Australia. Data are available from the Curtin University Human Research Ethics Committee (HREC) contact via hrec@curtin.edu.au for researchers who meet the criteria for access to confidential data.

**Funding:** This research was supported by Royal Life Saving Society Western Australia (WA) via an evaluation and research contract with Curtin University's Collaboration for Evidence, Research and Impact in Public Health (CERPIH) (JEL) (Contract Number 13834/RES-61536). URL: https://www.royallifesavingwa.com.au/ The funders supported data collection and the preparation of the manuscript.

**Competing interests:** The authors have declared that no competing interests exist.

drowning deaths occurring in young people under 24 years of age [3]. Risk factors for young people drowning vary by geographic location, for example, in high income country settings, gender, alcohol, supervision, proximity to water and swimming ability are well established risk factors together with social, environmental and structural factors [4]. In low and middle income country settings factors for young people may also include access to bodies of water, age and carrying capacity of the boat used for transport to and from work and school, weather and summer season [5].

In Australia a confluence of factors related to extensive coastline and waterways across Australia, the popularity of aquatic activities in this age group, and associated risk practices [1, 2, 4, 6–8] means young people are over-represented in water-related injury statistics and drowning deaths, a trend also seen internationally [1, 9–11]. Participation in recreational activities in and around beaches, rivers and waterways is often perceived to be part and parcel of an 'Australian way of life' [12, 13], and can be seen as a positive indicator of health and wellbeing, though continues to have fatal consequences, especially for males [7, 12, 14]. For example, in Australia in 2020/2021, there was a 21% increase in drowning deaths amongst young people aged 15–24 years; 77% of these deaths were males [9]. Deaths most frequently occurred in rivers and creeks, off rocks and at the beach [9].

Preventing young people from drowning is a multi-factorial issue [15]. Causes vary by individual (factors such as swimming skills [16] and personality traits including tendencies towards sensation-seeking) [14], environment (factors such as the aquatic setting and activity undertaken on the water) [15, 17] and social influence (factors such as peer norms) [18–21]. From a demographic perspective, young people and males are at increased risk of drowning [6], and more likely to use alcohol in the aquatic environment [2, 6, 9, 14, 15, 21]. In many countries, including Australia alcohol use around water is a risk factor for fatal and non-fatal drowning events especially amongst young adults [12]. However, these factors alone do not fully explain higher drowning rates for young adults. Accordingly, there has been a recent strategic focus on addressing a range of risk-taking behaviours underpinning these factors and associated behaviours in young people [2, 22].

Risk-taking is complex, dynamic and based on situated rationalities [23]. Recent research highlights the drowning prevention evidence has mostly focussed on males and children, as opposed to young adults [6, 13]. It has been established male drowning risk and risk perception are often associated with intentional activities described as 'fun' and recreational, often involving friends, fishing, jumping into water and boating [6, 14]. The literature suggests that young people, can underestimate the risk associated with aquatic activities and overestimate their ability to cope with that risk, partly explaining the higher rates of drowning amongst males [13, 15]. Research suggests that risk perception may play a protective role, motivating safer behaviours [14, 24]. However, research also suggests that young men may seek to protect a valued self or social identity through risk practices such as sensation-seeking, that may contribute to their sense of self-worth and build and sustain social networks and capital [20, 25]. Others' views, normative comparisons, social networks and groups, and individuals' audiences all influence risk practices among young people [16, 18, 26]. The complex relationship between social and physical desirability of participation in activities in and around waterways, higher drowning rates for males, propensity for risk, and the meaning young adults attach to risk locations and practices present unique challenges for drowning prevention research [14]. Greater understanding of the relationship between young adults and peer influence, decision making and risk perception may provide further insights into risk practices and account for different risk trajectories [27]. Accordingly, such information focussed on young adults in general, versus males specifically is critical for more effective interventions, particularly those that may benefit from a focus on settings, networks and key opinion leaders [28, 29].

The relative role of individual and sociocultural factors contributing to drowning risk for Australian adolescents and young adults is complex and poorly understood. Further exploration of the interaction of these factors and their influence on behaviour in and around water is warranted. This study aimed to examine the relationship between behaviour in and around water and: 1) alcohol consumption; 2) resistance to peer influence; 3) sensation-seeking; and 4) perception of risk among young people aged 15–24 years in Western Australia. Further, the research sought to examine potential predictors of these four factors separately and describe the extent to which they are associated with behavioural changes in and around water.

## Materials and methods

### Setting, survey participants, sample selection and recruitment

This study is part of a larger project evaluating an Australian youth water safety drowning prevention program. This research was undertaken in Perth, the state capital of Western Australia (WA) on the west coast of Australia. WA is Australia's largest geographical state with a population of 2,660,026 mainly residing along the coast [30]. The southwest coastal area (where Perth is located) has a temperate climate and summer occurs between the months of December–March [31]. A cross-sectional online survey was conducted at three time points (October 2019: T1; February 2020: T2 and March 2021: T3) with a convenience sample of young people. The criteria for inclusion were: English speaking, aged 15–24 years and residing in WA. The sampling protocol required 20% of participants to be from regional or remote WA, and 40% of the sample to be aged 15–19 years and 60% to be aged 20–24 years. Eligible participants were invited to complete an online survey using Qualtrics survey software (2017). An information sheet provided background to the study and the option to withdraw from the study at any point. Participants under 18 years were approved for mature minor informed consent by the University Human Research Ethics Committee (HREC). All participants were asked to indicate consent via an online question before they were able to proceed to the survey. The survey was distributed via networks of the WA peak, non-government drowning prevention organisation. Written informed consent was obtained electronically from all participants included in the study. The study was approved by the University Human Ethics Committee (Approval no. HR201/2014).

### Survey measures

The larger study uses a 53-item online survey. For the current analyses, a subset of data were derived from self-reported responses. The variables used in this study are described below and divided into *Behaviour*, *Factors affecting behaviour*, *Perception of risk and behaviour*.

**Behaviour.** Swimming after drinking alcohol. A single statement was used to measure water-related risk behaviour: "Have you swum after consuming alcohol?" [32]. Response options were Never, Sometimes and Always. The responses were dichotomised into 'Yes' (sometimes or always) and 'No' (Never).

**Factors affecting behaviour.** *Alcohol consumption*. The Alcohol Use Disorder Identification Test Consumption (AUDIT_C) [33] score assessed alcohol consumption. Three questions were included: 1) Frequency—How often do you have a drink containing alcohol? Response categories: Never; Monthly or less; 2–4 times a month; 2–3 times a week; 4 or more times a week; 2) Amount—How many standard drinks containing alcohol do you have on a typical day (when you are drinking alcohol)? Response categories: 1 or 2; 3 or 4; 5 or 6; 7 or 9; 10 or more; and 3) Frequency of high consumption—*How often do you have six or more standard drinks on one occasion*? Response categories: Never; Less than monthly; Monthly; Weekly;

Daily or almost daily. Scores for each question ranged from 0 to 4 points and total scores ranged from 0–12 points, with higher scores equating to higher consumption rates.

*Resistance to peer influence.* Participants were asked based on the Resistance to Peer Influence scale (RPI) [34] (a scale is a composite measure that is composed of several items that have a logical structure among them) [35]. Participants were asked "*How true are the following statements about you*?" and eight statements, including "*I think it's more important to be who I am than to fit in with the crowd*" and "*I would do something I know is wrong just to stay on my friends' good side*" (S1 Table for all eight statements). Each statement was coded as not true at all (1) to very true (4). Three items (1, 5 & 7) were reverse coded so that higher scores reflect greater susceptibility to peer influence. Mean Resistance to Peer Influence score (range from 1 through 4) were determined by averaging the individual statement scores.

*Sensation-seeking.* The Brief Sensation Seeking Scale (BSSS-4) [36] was used and participants were asked "*How much do you agree or disagree*" with each of the following statements: *1). I would like to explore strange places* (identifying thrill and adventure-seeking)*; 2). I like to do frightening things* (experience-seeking)*; 3). I like new and exciting experiences, even if I have to break the rules* (disinhibition); and *4). I prefer friends who are exciting and unpredictable* (boredom susceptibility)." The response scales were five-point Likert scales which ranged from 'strongly agree = 5' to 'strongly disagree = 1'. Higher scores correspond with a higher level of sensation-seeking. Mean sensation-seeking scores (range from 1 through 5) were calculated by averaging the individual response for the four statements.

**Perception of risk.** Participants rated their perception of risk using nine of the 14 items identified in the Benthin's Perception of Risk Scale [37]. The response used a seven-point Likert scale which varied for each of the nine items: 1) personal risk; 2) risk to peers; 3) benefit vs risk; 4) seriousness of effect; 5) information value; 6) perceived control; 7) peer influence; 8) admiration; and 9) avoidance. The perception of risk scores was calculated by averaging the individual response (1 through 7) for each item (see S2 Table for the full description of each of the nine items and their scales).

**Demographics.** The following demographic variables were collected: age, gender, location (by postcode used to categorise place of residence into two categories either Metropolitan (postcode 6000–6210) or Regional (postcode 6211–6999) WA), and current education level.

**Analysis.** Data were cleaned and participants with missing variables were removed (n = 312). A final sample of data (n = 730) was achieved (T1 (n = 374), T2 (n = 39) and T3 (n = 317)). Descriptive statistics summarised the participant demographic characteristics. Initially, a set of discrete bi-variate analyses were conducted for the variable–*Behaviour* (Have swum after drinking alcohol or Never swum after drinking alcohol) with the demographic variables (gender, age, place of residence, and education level) and each of the following four factors: AUDIT_C—alcohol consumption; sensation-seeking; resistance to peer influence; and perception of risk (personal risk, risk to peers, benefit versus risk, seriousness of effect, information value, personal control, peer influence, admiration, and avoidance). Associations were determined using Pearson chi-squared tests and t-tests. *P*-values <0.05 were considered statistically significant.

Separate forced entry binary logistic regression models then explored the influence of potential predictor variables on the dichotomous outcome: Behaviour *(0 = Never swum after drinking alcohol 1 = Have swum after drinking alcohol)*. Only the variables that were significant in the Pearson chi-squared analyses were included in the final model. Assumption testing conducted before the analysis did not indicate violations of multi-collinearity, outliers, or logit linearity. At each step, non-significant variables were removed from the model. A *p*-value threshold of 0.05 was used to limit the total number of variables included in the final model. The first step assessed the null model of the overall probability of the behaviour (swimming

whilst consuming alcohol) without adjustment for covariates. The second step included demographic variables age, gender and education. The third step included the three factors deemed likely to affect behaviour (AUDIT_C- alcohol consumption, resistance to peer influence and sensation-seeking). Whilst individually, gender, education, resistance to peer influence and risk perception involving *personal risk*, *risk to peers*, *benefit versus risk*, *information value*, *personal control*, *admiration and avoidance* showed an association with behaviour (Table 2), they were not significant when added to the model. Finally, the nine risk perception items were included in the model. The final logistic regression identified the impact of age, together with the predictor variables AUDIT_C–*alcohol consumption*, *sensation-seeking*; *peer influence* and *seriousness of effect* (risk perception items) on predicting swimming after drinking alcohol. The final model was statistically significant $\kappa^2$ *(df = 5, N = 623) = 242.625, p<0.001*, Cox and Snell *($R^2$ = 0.28)*, Nagelkerke *($R^2$ = 0.32)*. The model was 74.8% accurate in its predictions of behaviour (participating in water-based activity whilst consuming alcohol). Hosmer and Lemeshow test confirmed that the model was a good fit for the data $\kappa^2$ *(df = 8, N = 623) = 4.394, p = 0.820*. Significance levels are reported if p-values <0.05. Analyses were completed using SPSS (SPSS version 26) [38].

## Results

The majority of the participants (n = 730) were female (n = 537, 74.5%), lived in metropolitan Perth (n = 616, 84.4%) and attended university (n = 410, 56.9%). The mean age of participants was 19.9 years (SD = 2.13). Significant associations were found when behavioural responses (those who reported they had swum after drinking alcohol compared with those that had not) were analysed by age (p<0.001), gender (p = 0.021) and current education (p = 0.001). Demographic characteristics and the association with the outcome variable of interest, 'Behaviour' are shown in Table 1.

Analysis of behavioural responses by the four predictor variables also revealed significant associations across AUDIT_C—alcohol consumption, resistance to peer influence and sensation-seeking (p<0.001), and in all nine risk perception items (Table 2).

**Table 1. Demographics and association with the outcome variable 'Behaviour'.**

| Demographic characteristics | | Behaviour | |
|---|---|---|---|
| | | Have swum after drinking alcohol n (%) | Never swum after drinking alcohol n (%) |
| | p-value | | |
| **ALL** n (%) (n = 730) | | 294 (36.5) | 436 (54.2) |
| **Age** M (SD) | <0.001 | 20.54 (2.02) | 19.48 (2.1) |
| **Gender** n (%) (n = 721) | 0.021 | | |
| Male | | 88 (47.8) | 96 (52.2) |
| Female | | 205 (38.2) | 332 (61.8) |
| **Place of residence** n (%) (n = 730) | NS | | |
| Metropolitan | | 240 (39.0) | 376 (61.0) |
| Regional | | 54 (47.4) | 60 (52.6) |
| **Current education** n (%) (n = 721) | 0.001 | | |
| Not currently studying | | 98 (43.0) | 130 (57.0) |
| High school | | 7 (13.5) | 45 (86.5) |
| TAFE | | 11 (35.5) | 20 (64.5) |
| University | | 170 (41.5) | 240 (58.5) |

n-number; M–mean; SD- standard deviation; NS- Not Significant; TAFE–Tertiary and Further Education

**Table 2. Bivariate analyses of behaviour and factors affecting behaviour (AUDIT-C- alcohol consumption, resistance to peer influence, sensation-seeking and risk perception).**

| Predictor variables | p-value | Behaviour | |
|---|---|---|---|
| | | Have swum after drinking alcohol | Never swum after drinking alcohol |
| **ALL** n (%) | | 294 (36.5) | 436 (54.2) |
| **FACTORS AFFECTING BEHAVIOUR** M (SD) | | | |
| AUDIT_C—Alcohol consumption | p<0.001 | 5.52 (2.17) | 3.50 (1.93) |
| Resistance to Peer Influence | p<0.001 | 1.96 (0.46) | 1.81 (0.43) |
| Sensation-seeking | p<0.001 | 3.54 (0.76) | 3.26 (0.75) |
| **RISK PERCEPTION** M (SD) | | | |
| Peer Influence | p<0.001 | 3.78 (1.85) | 2.28 (1.55) |
| Seriousness of effect | p<0.001 | 5.63 (1.33) | 6.02 (1.18) |
| Personal risk | p<0.001 | 3.16 (1.71) | 2.35 (1.67) |
| Risk to peers | p<0.001 | 2.40 (1.36) | 1.84 (1.31) |
| Benefit vs risk | p<0.001 | 2.27 (1.46) | 1.65 (1.20) |
| Information value | p<0.001 | 5.62 (1.42) | 6.03 (1.23) |
| Personal control | 0.002 | 3.89 (1.70) | 3.47 (1.90) |
| Admiration | 0.017 | 4.52 (1.58) | 4.22 (1.78) |
| Avoidance | 0.004 | 5.44 (1.60) | 5.78 (1.44) |

n-number; M–mean; SD- standard deviation

Table 3 presents the association between participation in swimming after drinking alcohol and age, AUDIT_C—alcohol consumption, sensation-seeking and risk perception (peer influence and seriousness of effect) after simultaneous adjustment for these variables. Participants were almost 50% more likely to swim after drinking alcohol with every year they got older (OR 1.46 95% CI 1.32–4.63). Similarly, participants were almost 40% more likely to swim after drinking alcohol with increasing sensation seeking and peer-influence scores (OR 1.46 95% CI 1.30–1.64 and OR 1.44 95% CI 1.09–1.89 per 1-unit increase in their respective scores). For every 1-unit increase in the AUDIT-C score participants were 60% more likely to swim after drinking alcohol (OR 95% CI 1.60 1.44–1.78). In contrast, those who more strongly considered the seriousness of an adverse event were 15% less likely to have swum after drinking alcohol (OR 0.85 95% CI 0.73–0.99 per 1-unit increase in the score).

**Table 3. Predictor coefficients for the logistic regression model predicting participation in swimming after drinking alcohol.**

| Participant characteristic | Behaviour model–Swimming after drinking alcohol | |
|---|---|---|
| | OR (95% CI) | p |
| Age | 1.46 (1.32–4.63) | <0.001 |
| Alcohol consumption | 1.60 (1.44–1.78) | <0.001 |
| Sensation-seeking | 1.44 (1.09–1.89) | 0.010 |
| **Risk perception** | | |
| Peer Influence | 1.46 (1.30–1.64) | <0.001 |
| Seriousness of effects | 0.85 (0.73–0.99) | 0.045 |

CI—confidence intervals; SE—standard error

## Discussion

We set out to explore the complex interaction of factors influencing the behaviour of young Western Australians aged 15–24 years (alcohol consumption, perception of risk, sensation-seeking and peer influence) in and around waterways. For many young Australians, alcohol and risk-taking is an inherent part of identity formation [39] occurring in a pervasive alcogenic environment [12, 40, 41]. An 'aquatic alcogenic environment' described in previous research [12], highlights the very commonplace practice of alcohol consumption in and around water. The findings in this study are novel and extend a small but growing body of literature that examines the complex socio-cultural relationships in and around waterways, and the meaning that both young male and female adults attach to these factors.

The findings in this study suggest that peer influence and sensation-seeking influenced swimming after drinking alcohol, a practice that those more likely to swim and drink felt would be admired by their peers. These findings are consistent with another Australian study that found young males who have positive attitudes towards drinking and swimming, consider their peers to hold similar attitudes towards drinking and swimming and would perform the action [20]. Of interest, in this study young females were over-represented providing an interesting and previously untapped insight into the female perspective on the role of peers and sensation seeking. Participants in our study who had swum after drinking alcohol reported more personal control over risk, less fear of risk to themselves and others, and less seriousness of effects, suggesting those who sensation seek or frequently drink may have a lower perceived susceptibility to drowning, a finding consistent with previous research [12, 20, 42]. Conversely, those who had never swum after drinking alcohol were more likely to consider the seriousness of an adverse event, and reported higher risk to both themselves and others, greater risks than benefits, and less personal control over risk. These contrasting behaviours amongst those who *do* drink and swim and those who *do not* have direct implications for the design and delivery of future drowning prevention interventions for young adults.

Our research reinforces previous findings that sensation-seeking increases when alcohol is consumed. It is well established that sensation-seeking, coupled with group norms are factors that may facilitate and amplify risk practices in young people, including those related to alcohol consumption [6, 21, 42–44]. Previous research has noted that males, specifically those who were younger and who scored highly on sensation-seeking tended to mix with peers who drink more frequently [20, 42, 43]; and may have lower self-efficacy around decision making [43]. Moran, 2011 describes 'dangerous masculinities' whereby aquatic recreational activity (fishing and surfing) is a masculinised and gendered pursuit, a notion that appears to be firmly entrenched in the social norms of young men. A recent review describes an Australian study that found females are also engaging in high-risk behaviours and activities in aquatic locations similar to males, especially when drinking around waterways [6, 44]. These are important insights for awareness raising and education endeavours targeting younger adults. Lupton & Tulloch [18] describe risk across a life-course trajectory that involves significant risk-taking by young people. Compared with their older peers, younger adults may have had less exposure to risk environments or to opportunities to build their skills and self-efficacy to reduce risks that may be inherent in sensation-seeking activities [42]. Accordingly, this research reinforces the opportunity to exploit and target peer group norms (e.g., peer education), skills and self-efficacy (e.g., assertiveness training) [42, 43] as part of the suite of strategies delivered to younger adults. The findings also suggest the need for segmented prevention messages that account for the differing peer roles that young people hold within their social networks given the moderating effect of social influence [28]. Like recommendations by Abercromby et al (2020) [12], Calverley et al (2020) [42], and Hamilton et al 2022 [21] our findings highlight the need for better

understanding regarding the role of risk-taking and risk-averse peers in drowning prevention efforts. Identifying the wide range of reasons that young people participate in risk practices would be instructive in developing more targeted prevention measures for Australian and international drowning prevention practitioners and policy makers. This includes an awareness of the positive associations with risk and sensation-seeking and their role in achieving desired social outcomes including social capital, reputational and experiential gains [25].

A serious side effect or something bad happening was more likely to be front of mind for those who did not drink and swim in our study. This result is interesting considering recent findings from WA research with young people who dismissed media messages that emphasised the negative and serious consequences of drinking alcohol [12]. Hamilton and colleagues [20] found strategies that aim to increase the negative consequences (e.g. increasing the chance of injury/accidents) of drinking and swimming may be effective. However, the effectiveness of threatening messages appears dependent on the individual considering themselves vulnerable to the threat portrayed [20, 45], regardless of the severity of the risks involved. Previously it has been found that whilst young people are aware of the serious consequences of excessive alcohol consumption, they consider these to be the cost of perceived benefits [12]. Of interest is the role of the specific setting, with boats and boating quoted as the sites of serious injury, that had sustained and significant impacts and outcomes which included death [12]. Recent research suggests the use of passive messaging using to place drowning prevention top of mind [13]. For example, in the New Zealand Swim Reaper campaign, young people are specifically reached via geo-targeted messages using Instagram and other social channels. Safety signage is also placed at selected locations where drownings occur [46]. The targeted placement of media messages using digital technology together with environmental cues are vital to highlight the consequences of alcohol use during aquatic activities [13, 21].

However, it has also been suggested that whilst young Australian adults are aware of the seriousness of mixing alcohol and aquatic activities, their inexperience, short-term focus and impulsivity allows them to disassociate themselves from the possibility of experiencing any serious consequences for themselves or others [42, 47] which may be amplified by feelings of situational disinhibition [48]. Risk, in the context of alcohol, water and young people, may be seen as an important cultural practice which is reinforced by peers and the environment. Any interventions that are serious about tackling risk practices will need to confront this reality in their design and delivery.

We found alcohol consumption was significantly associated with activities in and around the water; not unexpected in an Australian state surrounded by almost 13,000 kilometres of coastline [49]. These findings are concerning given alcohol contributes to around 20% of drowning fatalities in WA [50]. Recent qualitative research [12] has indicated that young Australians regularly mix alcohol and activities around waterways, citing it as a cultural norm, despite acknowledging and knowing the risks [42]. Consequently strategies that focus only on awareness or knowledge about risk are unlikely to resonate with the target audience [51]. Our research strengthens the call for comprehensive interventions that not only encourage young people to consider the risks associated with drinking and swimming, and the seriousness of the effects of drinking and swimming, but provide environmental supports that seek to mitigate the pervasive effects of the aquatic alcogenic environment. Described earlier, where alcohol advertising widely promotes drinking in and around the water, and is ineffactually regulated [41, 52–54]. Increased advocacy efforts are required to remove alcohol advertising featuring water-based activities and restrict alcohol outlets near waterways as a key public health strategy in drowning prevention. In particular, this means highlighting and advocating for government intervention to counter the self-regulating nature of the Australian Alcohol

Beverages Advertising Code and the relatively weak placement rules for alcohol advertising [52, 53, 55]. As has been suggested in the literature, it is doubtful that this strategy will have any material impact on alcohol marketing exposure to young people [55–57]. The focus has to include tackling the commercial determinants of health and the public health consequences arising from for-profit entities such as the alcohol industry, their activities, and the social structures that sustain them [58]. As Crawford and colleagues [59] have recently argued in relation to government intervention, that while less intrusive measures are usually the most acceptable to individuals, they are usually also the least effective. In other areas of health there is majority support for government intervention, particularly when it relates to children and young people and the impact of the commercial determinants on health.

### Limitations and strengths

Our study had several limitations. First, data collection used the existing networks of the peak drowning prevention agency in one Australian state, consequently, the findings may not reflect the general young adult population. This study was undertaken in WA and care should be taken when extrapolating findings to other locations. Data were self-reported and subject to recall and social desirability bias. WA COVID 19 border shutdowns and social distancing policy in March-April, 2021 impacted recruitment and data collection at T2. Females, and those who lived in the metropolitan area were over-represented in the final sample, and more than one-half of participants had attended a tertiary institution. The use of a single item for behaviour item analysed swimming after consuming alcohol, limits generalisability of the findings. The strengths of this study include the novel use of predictor variables to examine behaviour in and around waterways. To the best of the authors' knowledge, this is the first study to examine the specific predictor variables in the drowning prevention literature; and the findings contribute to a small but growing evidence base examining the individual and sociocultural factors contributing to drowning risk for young adults.

### Conclusion

This study adds to and extends the small body of literature that examines risk and risk perception in young people as important predictors for a range of practices in and around waterways in high-income countries. The significant influence of alcohol, sensation seeking, peers and perceived seriousness of injury are important but underutilised considerations in the development of prevention programs designed to discourage alcohol use in, on, or around waterways. Our novel findings that include both a male and female perspective should be used to improve the awareness and education components of young adult water safety strategies with the addition of supporting structural and environmental strategies to reduce the significant physical, social, and economic burden associated with drowning in Australia and other locations with large coastlines such as Canada and New Zealand.

### Supporting information

**S1 Table. Resistance to Peer Influence scale (RPI).**
(TIF)

**S2 Table. Benthin's perception of risk scale.**
(TIF)

## Acknowledgments

We would like to acknowledge the young people of Western Australia who consent to being part of our research endeavours and the Royal Life Saving Society of Western Australia (RLSSWA) for their support in data collection and manuscript preparation.

## Author Contributions

**Conceptualization:** J. E. Leavy, M. Della Bona, G. Crawford.

**Formal analysis:** J. E. Leavy, M. Della Bona, G. Crawford.

**Funding acquisition:** J. E. Leavy.

**Methodology:** J. E. Leavy, G. Crawford.

**Writing – original draft:** J. E. Leavy, M. Della Bona, M. Abercromby, G. Crawford.

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
