## [Decision Letter · Decision Letter 0]

22 Aug 2022

PONE-D-22-18336Drinking and swimming around waterways: the role of alcohol, sensation-seeking, peer influence and risk in young people.PLOS ONE

Dear Dr. Leavy,

Thank you for submitting your manuscript to PLOS ONE. After careful consideration, we feel that it has merit but does not fully meet PLOS ONE’s publication criteria as it currently stands. Therefore, we invite you to submit a revised version of the manuscript that addresses the points raised during the review process.

Editor comments:1) I understand that you cannot share raw data but you will need to update the data sharing statement so that it fulfils the criteria as suggested. you need to let people know how they could access the data if they really wanted to. For example, data cannot be shared publicly because of XXX. Data are available from XXX (contact via XXX) for researchers who meet the criteria. General comments:This paper provides a new contribution to understanding behaviours and perceptions of alcohol consumption and swimming within young people in Western Australia. The tools used are appropriate and interpretation is meaningful. Some concerns around interpretation and context development provided using young men but the sample population tested is predominantly females. Also manuscript could benefit from broader international relevance, presenting WA as a case study, and relate back to national and international scope. Some minor comments are provided below. Abstract:Line 43: meaning young men attach to risk locations - not sure this can be part of your interpretation when most of your sample were female - suggest change to young adults?Introduction:General: much of the context is framed around young males, and yes they are a high risk demographic, your research would be stronger if it were framed more about young adults given the sample. Or even highlight that most research has been on young men - as opposed to young adults in general. I also think for an international scope, a broader context would be more beneficial before going into Australia and then WA. Also need to highlight that WA may not be representative of Australia etc..   Line 58: and at the beach (grammatical error) Methods:General:  Are the headings starting each paragraph?? i.e. Alcohol consumption. (line 123) may be better as a sub-heading?  Similar for resistance to peer influence and sensation seeking.Behaviour Scales would benefit from a brief overview of what they measure for readers who are not familiar with psychology.Line 112: Ethics approval numbers need to be added.Line 114: Subset (remove hyphen)Line 120: Why were sometimes and always combined? Line 157: Can you describe the score determination and direction of the score as you have for previous scales i.e. high score means greater risk perception??Line 164: Were there any collinearity issues across the three time points? Were responses similar in breadth etc. I can imagine that there may have been some diffferences between the 2019/20 cohort to the 2021 cohort? How did you account for this given the challenges experienced over this timeframe i.e. pandemic, bushfires etc... Line 174 onwards: Why select multiple separate forced binary logistic regressions? Were there benefits to this over say a multiple/hierachical/stepwise logistic regression analyses?  Results:Please report data in line with PLOSONE reporting guidelines https://journals.plos.org/plosone/s/submission-guidelines#loc-statistical-reporting . You cannot present 0 as a p-value. Please change to p<0.001. Please rectify this in the results text and the tables.    Discussion: The discussion seems to show how the results support previous research. However this missed the opportunity to demonstrate how the results extend what the previous research found. Again much focus on male behaviours and perceptions but your study is more than just men. In fact it is more the opposite thanks to your sample. I think your findings would be more novel if it capitalised on this, and showed how it could benefit Australia/international drowning practitioners.   Lines 280-300: This paragraph/s is/are great but, especially since current media is dismissed and seriousness of an event is not front of mind for participants who drank, i think it could be improved with the proposal of alternative messaging approaches that may differ e.g. swim reaper (Water Safety New Zealan), passive education at less chaotic developmental times (e.g. early childhood/primary school; see Lawes et al 2020 Risky business) , etc. For example how should intervention design and delivery differ in the context of these results.  Similarly, i have found mostly that, in an australian context alcohol is not just a young adult issue - it spans all ages... so effective messaging is so important and impact can be far reaching.  Line 306: waterways (needs an s)Line 311: do you mean mitigate instead of ameliorate?   ==============================

We look forward to receiving your revised manuscript.

Kind regards,

Jasmin C. Lawes, Ph.D.

Academic Editor

PLOS ONE

Journal Requirements:

a) Did participants provide their written or verbal informed consent to participate in this study?

Reviewers' comments:

Reviewer's Responses to Questions

**Comments to the Author**

1. Is the manuscript technically sound, and do the data support the conclusions?

Reviewer #1: Yes

Reviewer #2: Yes

2. Has the statistical analysis been performed appropriately and rigorously? 

Reviewer #1: Yes

Reviewer #2: Yes

3. Have the authors made all data underlying the findings in their manuscript fully available?

Reviewer #1: Yes

Reviewer #2: No

4. Is the manuscript presented in an intelligible fashion and written in standard English?

Reviewer #1: Yes

Reviewer #2: Yes

5. Review Comments to the Author

Reviewer #1: this manuscript aims to bring light to an important and still poorly understood issue regarding youth safety around water. I congratulate the authors for studying this subject in a very robust way providing evidence to base further decisions on intervention to prevent drowning in this age group.

Reviewer #2: Thanks for the opportunity to review Manuscript ID PONE-D-22-18336 entitled “Drinking and swimming around waterways: the role of alcohol, sensation-seeking,

peer influence and risk in young people” which was submitted for potential publication in PLOS ONE. This study uses a survey to explore young adults and their relationship with alcohol, peer influence, sensation seeking and perception of risk around waterways. This study is novel and a valuable contribution to the literature. It is well written but I have some specific suggestions for minor revisions. I look forward to seeing the publication in print in due course.

Abstract

Line 35 – incomplete sentence starting with – the final sample

Its such a heavily female sample, unsure about the specific reference to males in line 43

Introduction

Generally well written and referenced however, I didn’t see this study cited – it may be of relevance for the introduction and also the discussion - https://www.tandfonline.com/doi/full/10.1080/00049530.2022.2029221

Materials and methods

In the first section might be worth describing WA in more detail as PLOSONE is an international journal with an international audience

Survey measures – survey tool provided as supplementary?

Analysis – is there a brief explanation regarding why the T2 sample is so low compared to the others?

Results

Line 203 and again at line 210 and Table 2 and Table 3 – should p=0.000 for age not be reported as p<0.001?

Discussion

Line 235 – check and correct reference format here

Line 272 and 273 – needs references to named studies?

Line 306 – waterways not waterway

Strengths and limitations section – doesn’t appear to have any strengths? Perhaps the novel approach is a strength, if so expand upon this as currently written to make it clearer. It’s a novel combination of measures used in the survey so would encourage authors to highlight this!

Limitations – very female (mentioned), check if these are also adequately addressed. metro and highly educated sample. Were there covid or some other impacts on recruitment, especially T2?

Conclusions – no comments.

6. PLOS authors have the option to publish the peer review history of their article (what does this mean?). If published, this will include your full peer review and any attached files.

Reviewer #1: No

Reviewer #2: No

---

## [Author Response · Author response to Decision Letter 0]

21 Sep 2022

21092022

We have added the statement in Methods indicating written informed consent was obtained from the participants included in the study see line: 129-130. Thank-you

Response to reviewer comments

Thank-for to the Editor and both reviewers for your thoughtful comments and feedback. Each comment has been addressed below and in the attached updated manuscript (highlighted in yellow in the updated version). We look forward to your reply in due course.

Editor comments:

Comment 1:

1) I understand that you cannot share raw data but you will need to update the data sharing statement so that it fulfils the criteria as suggested. you need to let people know how they could access the data if they really wanted to. For example, data cannot be shared publicly because of XXX. Data are available from XXX (contact via XXX) for researchers who meet the criteria. 

Reply 1: Thank-you. This has been amended and updated as suggested by the Editor in the submission portal.

General comments:

This paper provides a new contribution to understanding behaviours and perceptions of alcohol consumption and swimming within young people in Western Australia. The tools used are appropriate and interpretation is meaningful. Some concerns around interpretation and context development provided using young men but the sample population tested is predominantly females. Also manuscript could benefit from broader international relevance, presenting WA as a case study, and relate back to national and international scope. Some minor comments are provided below.

Response General comments:

Introduction: Paragraph 1 lines 49-55 an international context for drowning and young adults has been added to the opening paragraph. Please see the yellow highlighted text. 

Paragraph 4: lines 79-81 highlight a recent review which has found most research is focused on males and children and this is an area that needs to expand to young adults. Where appropriate, overuse of the word males has been removed, and young adult has been used to provide a more balanced perspective. Lines 98-99, as per your suggestion we have added that more research on young adults, in general, is required. 

Abstract:

Comment 1

Line 43: meaning young men attach to risk locations - not sure this can be part of your interpretation when most of your sample were female - suggest change to young adults?

Response 1: Thank-you this has been changed to young adults. See Abstract, line 43. 

Introduction:

Comment 2

General: much of the context is framed around young males, and yes they are a high risk demographic, your research would be stronger if it were framed more about young adults given the sample. Or even highlight that most research has been on young men - as opposed to young adults in general. 

Response 2: Thank-you we have changed the text to reflect those observations and added a recent review [1] which highlights a lack of research around females in HICs. See lines 79-81. 

Comment 3

3a: I also think for an international scope, a broader context would be more beneficial before going into Australia and then WA. 3b: Also need to highlight that WA may not be representative of Australia etc.. 

Response 3a: Please see above. Paragraph 1 lines 49-55 an international context for drowning and young adults has been added to the opening paragraph. Please see yellow highlighted text. 

Response 3b: Thank-you for the observation we had noted that in the limitations section. See lines 366-368, ‘First, data collection used the existing networks of the peak drowning prevention agency in one Australian state, consequently, the findings may not reflect the general young adult population’. 

Comment 4 

Line 58: and at the beach (grammatical error) 

Response 4: This has been corrected see line now 67.

Methods:

Comment 5

General: Are the headings starting each paragraph?? i.e. Alcohol consumption. (line 123) may be better as a sub-heading? Similar for resistance to peer influence and sensation seeking.

Response 5: We have made these changes as suggested.

Comment 6

Behaviour Scales would benefit from a brief overview of what they measure for readers who are not familiar with psychology.

Response 6: A brief description has been added at lines 153-155 it reads (a scale is a composite measure that is composed of several items that have a logical structure among them) [2].

Comment 7

Line 112: Ethics approval numbers need to be added.

Response 7: These have been added see line 130 HR201/2014. 

Comment 8

Line 114: Subset (remove hyphen)

Response 8: This has been removed. See subset – now line 132.

Comment 9

Line 120: Why were sometimes and always combined? 

Response 9: The responses for “always’ (n=13) and ‘sometimes’ (n=281) were deemed appropriate to combine as we were interested if participants had ever taken part in the behaviour or not. The variable was from an existing scale which was also two categories (yes/no) see Moran K., 2011 [3].

Comment 10

Line 157: Can you describe the score determination and direction of the score as you have for previous scales i.e. high score means greater risk perception??

Response 10: All nine items from the Benthin scale were used as individual items only (i.e. an overall risk perception score was not calculated). The scale used and how each of the items are scored is described in full in the supplementary table S2, whereby 1 through 7 varies depending on the specific item, therefore a generic description 1 = low and 2 = high in the methods will not be helpful, it is better to refer the reader to the table. Please see lines 177-179. “The perception of risk score was calculated by averaging the individual response (1 through 7) for each item (see S2 Table for the full description of each of the nine items and their scales).” We are happy to include the S2 table into the methods if the Editor would prefer.

Comment 11

Line 164: Were there any collinearity issues across the three time points? Were responses similar in breadth etc. I can imagine that there may have been some diffferences between the 2019/20 cohort to the 2021 cohort? How did you account for this given the challenges experienced over this timeframe i.e. pandemic, bushfires etc... 

Response 11: Thank you for the questions. The preliminary analysis confirmed there were no significant differences in behaviour by timepoint, thus, timepoints were combined for this paper. Whilst the response rate at T2 is smaller (potentially due to a short, one-off lockdown imposed in WA) responses were not significantly different from responses obtained at T1 or T3. Fortunately, WA did not experience the events seen on the east coast, i.e. bushfires, flooding and extended lockdowns. 

Comment 12

Line 174 onwards: Why select multiple separate forced binary logistic regressions? Were there benefits to this over say a multiple/hierachical/stepwise logistic regression analyses? 

 Response 12: During preliminary analysis, several approaches to the logic regression were tested, including multiple and hierarchical logistic regression. Forced logistic regression provided the strongest and more refined predictive model.

Results:

Comment 13

Please report data in line with PLOSONE reporting guidelines https://journals.plos.org/plosone/s/submission-guidelines#loc-statistical-reporting . You cannot present 0 as a p-value. Please change to p<0.001. Please rectify this in the results text and the tables. 

Response 13: Thank you for noting this. It has been updated in the text and tables. See lines 226, and 233 and Tables 1 and 2. 

Discussion: 

Comment 14

The discussion seems to show how the results support previous research. However this missed the opportunity to demonstrate how the results extend what the previous research found. Again much focus on male behaviours and perceptions but your study is more than just men. In fact it is more the opposite thanks to your sample. I think your findings would be more novel if it capitalised on this, and showed how it could benefit Australia/international drowning practitioners. 

Response 14: Thank you for the suggestion. We have used our findings together with the suggested references Lawes et al 2020 (14 in text) [4] and Hamilton et al 2022 (22 in text) [5] to add text to the Discussion to better describe how future drowning prevention interventions could benefit from these insights. See lines 297, 303 and 323, 328.

Specific changes: Discussion paragraph 1 – lines 261 to 263 additional text added that the study findings are novel as it includes both a male and female perspective. Now reads ‘The findings in this study are novel, and extend a small but growing body of literature that examines the complex socio-cultural relationships in and around waterways, and the meaning that both young male and female adults attach to these factors’.

Paragraph 2 –females were overrepresented in the study, we have added a recent systematic review and Australian study that found females were increasing their risk taking when consuming alcohol around waterways to complement the male perspective in this paragraph. See lines 269 to 271 ‘Of interest, in this study young females were over-represented providing an interesting and previously untapped insight into the female perspective on the role of peers and sensation seeking’ and Paragraph 3 lines 290- 291 ‘Females are also engaging in high-risk behaviours and activities in aquatic locations similar to males, such as ingesting alcohol in aquatic locations [1, 6]’

Comment 15

Lines 280-300: This paragraph/s is/are great but, especially since current media is dismissed and seriousness of an event is not front of mind for participants who drank, i think it could be improved with the proposal of alternative messaging approaches that may differ e.g. swim reaper (Water Safety New Zealan), passive education at less chaotic developmental times (e.g. early childhood/primary school; see Lawes et al 2020 Risky business), etc. For example how should intervention design and delivery differ in the context of these results. 

 Response 15: Thank-you for these suggestions. We have added your reference Lawes et al. 2020 [4] to both the introduction and the discussion. Please see lines 322 – 328 for additional text to address alternative messaging including environmental cues and the Swim Reaper NZ campaign as a novel and appropriate approach. It now reads “Recent research suggests the use of passive messaging using to place drowning prevention top of mind (14). For example, in the New Zealand Swim Reaper campaign, young people are specifically reached via geo-targeted messages using Instagram and other social channels. Safety signage is also placed at selected locations where drownings occur (47). The targeted placement of media messages using digital technology together with environmental cues are vital to highlight the consequences of alcohol during aquatic activities (14, 22).” 

Comment 16

Similarly, i have found mostly that, in an australian context alcohol is not just a young adult issue - it spans all ages... so effective messaging is so important and impact can be far reaching. 

 Response 16: Thank-you for your insights.

Comment 17

Line 306: waterways (needs an s)

Response 17: This has been corrected see line 342 now reads …waterways. 

Comment 18

Line 311: do you mean mitigate instead of ameliorate? 

Response 18: Thank-you for the pickup. Replaced with mitigate see line 348 now.

Reviewer #1: this manuscript aims to bring light to an important and still poorly understood issue regarding youth safety around water. I congratulate the authors for studying this subject in a very robust way providing evidence to base further decisions on intervention to prevent drowning in this age group.

Response: Thank-you for your comments on our paper, they are most appreciated. 

Reviewer #2: Thanks for the opportunity to review Manuscript ID PONE-D-22-18336 entitled “Drinking and swimming around waterways: the role of alcohol, sensation-seeking, peer influence and risk in young people” which was submitted for potential publication in PLOS ONE. This study uses a survey to explore young adults and their relationship with alcohol, peer influence, sensation seeking and perception of risk around waterways. This study is novel and a valuable contribution to the literature. It is well written but I have some specific suggestions for minor revisions. I look forward to seeing the publication in print in due course.

Response 1: Thank-you for your comments on our paper, please see our specific responses below. 

Abstract

Comment 2

Line 35 – incomplete sentence starting with – the final sample

Response 2a: Line 35-37 has been updated to read ‘ The final sample (n =730) participants, consisted of females (n=537, 74.5%), metropolitan dwelling (n=616, 84.4%) and attended university (n=410, 56.9%).

Its such a heavily female sample, unsure about the specific reference to males in line 43

Response 2b: We agree and is consistent with the Editor’s comment has been changed to young adults see line 43

Introduction

Comment 3

Generally well written and referenced however, I didn’t see this study cited – it may be of relevance for the introduction and also the discussion - https://www.tandfonline.com/doi/full/10.1080/00049530.2022.2029221

Response 3: Thank-you for the additional reference which we have used in the introduction line 71, 73 ( is ref 22 [5] in Introduction) lines 327-329 highlighting the need for digital and app based strategies to highlight the consequences of alcohol use around waterways. See (ref 22 also used in the Discussion)

Materials and methods

Comment 4

In the first section might be worth describing WA in more detail as PLOSONE is an international journal with an international audience

Response 4: Thank-you for your comment, we have added some Western Australian specific context lines 114 to 118 , see ‘This research was undertaken in Perth, the state capital of Western Australia (WA) on the west coast of Australia. WA is Australia’s largest geographical state with a population of 2,660,026 mainly residing along the coast [7]. The southwest coastal area (where Perth is located) has a temperate climate and summer occurs between the months of December–March [8]’.

Comment 5:

Survey measures – survey tool provided as supplementary?

Response 5: The authors may be contacted for a copy of the full survey. 

Comment 6:

Analysis – is there a brief explanation regarding why the T2 sample is so low compared to the others?

Response 6: T2 data collection commenced during the Western Australian COVID19 lockdown in March 2021. All borders remained closed for an extended period, social distancing impacted recruitment methods e.g. public outdoor events for young adults, Orientation days at universities and aquatic industry events were shut-down making recruitment very difficult. Now added as a limitation, see lines 370-371. 

Results

Comment 7

Line 203 and again at line 210 and Table 2 and Table 3 – should p=0.000 for age not be reported as p<0.001?

Response 7: Thank you for noting this. It has been updated and corrected in the text and the tables in the manuscript. Please see the comment above.

Discussion

Line 235 – check and correct reference format here Response: The Sinkinson, 2014 reference has been updated and is now ref 40 

Line 272 and 273 – needs references to named studies? Response: We have added the number however I am not 100% sure they are needed here. Refs 13 and 43 are now in the text. Now line 302 and we have also added Hamilton et al., 2022 to this sentence. 

Line 306 – waterways not waterway Response: Thank you the s has been added to waterways now at line 342 

Comment 8.

Strengths and limitations section – doesn’t appear to have any strengths? Perhaps the novel approach is a strength, if so expand upon this as currently written to make it clearer. It’s a novel combination of measures used in the survey so would encourage authors to highlight this!

Response 8: Thank you for your suggestion. This section has been rewritten and the strengths highlighted, see lines 375 – 379. The text now reads “The strengths of this study include the novel use of predictor variables to examine behaviour in and around waterways. To the best of the authors’ knowledge, this is the first study to examine the specific predictor variables in the drowning prevention literature; and the findings contribute to a small but growing evidence base examining the individual and sociocultural factors contributing to drowning risk for young adults” 

Comment 9:

Limitations – very female (mentioned), check if these are also adequately addressed. metro and highly educated sample. Response 9a: please see line we mention 371-372 where we mention metropolitan dwelling and university educated were over represented. 

Were there covid or some other impacts on recruitment, especially T2?

Response 9b: We have now added COVID 19 border shutdowns in March-April, 2021 as a limitation for data collection at T2. See lines 370-371. 

Conclusions – no comments.

Thank-you. 

References

1. Roberts K, Thom O, Devine S, Leggat PA, Peden AE, Franklin RC. A scoping review of female drowning: an underexplored issue in five high-income countries. BMC Public Health. 2021;21(1):1072. doi: 10.1186/s12889-021-10920-8.

2. Loewenthal KM, Lewis CA. What is a good psychological measure? An introduction to psychological tests and scales (3rd ed)2020. p. 1-22.

3. Moran K. (Young) Men behaving badly: dangerous masculinities and risk of drowning in aquatic leisure activities. Annals of Leisure Research. 2011;14(2-3):260-72.

4. Lawes JC, Ellis A, Daw S, Strasiotto L. Risky business: a 15-year analysis of fatal coastal drowning of young male adults in Australia. Injury Prevention. 2021;27(5):442. doi: 10.1136/injuryprev-2020-043969.

5. Hamilton K, Keech JJ, Willcox - Pidgeon S, Peden AE. An evaluation of a video-based intervention targeting alcohol consumption during aquatic activities. Australian Journal of Psychology. 2022;74(1):2029221. doi: 10.1080/00049530.2022.2029221.

6. Peden AE, Franklin RC, Leggat PA. Breathalysing and surveying river users in Australia to understand alcohol consumption and attitudes toward drowning risk. BMC Public Health. 2018;18(1):1-18.

7. Australian Bureau of Statistics. Western Australia 2021 All persons Quick Stats. Canberra2021.

8. Australian Bureau of Meterology. Nyoongar Weather Calendar. 2016.

---

## [Editor Report · Decision Letter 1]

10 Oct 2022

Drinking and swimming around waterways: the role of alcohol, sensation-seeking, peer influence and risk in young people.

PONE-D-22-18336R1

Dear Dr. Leavy,

We’re pleased to inform you that your manuscript has been judged scientifically suitable for publication and will be formally accepted for publication once it meets all outstanding technical requirements.

Kind regards,

Jasmin C. Lawes, Ph.D.

Academic Editor

PLOS ONE
---

## [Editor Report · Acceptance letter]

13 Oct 2022

PONE-D-22-18336R1 

Drinking and swimming around waterways: the role of alcohol, sensation-seeking, peer influence and risk in young people. 

Dear Dr. Leavy:

I'm pleased to inform you that your manuscript has been deemed suitable for publication in PLOS ONE. Congratulations! Your manuscript is now with our production department. 

Kind regards, 

on behalf of

Dr. Jasmin C. Lawes 

Academic Editor

PLOS ONE